# Association between Air Pollution and Squamous Cell Lung Cancer in South-Eastern Poland

**DOI:** 10.3390/ijerph191811598

**Published:** 2022-09-15

**Authors:** Jan Gawełko, Marek Cierpiał-Wolan, Second Bwanakare, Michalina Czarnota

**Affiliations:** 1Institute of Medical Sciences, College of Medical Sciences, University of Rzeszów, 35-959 Rzeszów, Poland; 2Statistical Office in Rzeszów, 35-959 Rzeszow, Poland; 3Institute of Economics and Finance, College of Social Sciences, University of Rzeszów, 35-959 Rzeszów, Poland; 4Institute of Economics and Finance, Faculty of Socio-Economics, Cardinal Stefan Wyszynski University, 01-938 Warsaw, Poland; 5Institute of Health Sciences, College of Medical Sciences, University of Rzeszów, 35-959 Rzeszów, Poland

**Keywords:** air pollutants, morbidity, squamous cell carcinoma, lung cancer, principal components analysis, econometrics

## Abstract

Air pollution is closely associated with the development of respiratory illness. The aim of the present study was to assess the relationship between long-term exposure to PM2.5, PM10, NO_2_, and SO_2_ pollution and the incidence of lung cancer in the squamous subtype in south-eastern Poland from the years 2004 to 2014. We collected data of 4237 patients with squamous cell lung cancer and the level of selected pollutants. To investigate the relationship between the level of concentrations of pollutants and the place of residence of patients with lung cancer in the squamous subtype, proprietary pollution maps were applied to the places of residence of patients. To analyze the data, the risk ratio was used as well as a number of statistical methods, i.e., the pollution model, inverse distance weighted interpolation, PCA, and ordered response model. Cancer in women and in men seems to depend in particular on the simultaneous inhalation of NO_2_ and PM10 (variable NO_2_PM10) and of NO_2_ and SO_2_ (variable NO_2_ SO_2_), respectively. Nitrogen dioxide exercises a synergistic leading effect, which once composed with the other elements it becomes more persistent in explaining higher odds in the appearance of cancers and could constitute the main cause of squamous cancer.

## 1. Introduction

Air pollution is one of the leading causes of morbidity and mortality worldwide [1,2]. It is estimated that prolonged exposure to ambient dust (PM) with a diameter less than or equal to 2.5 μm (PM2.5) resulted in 4 to 9 million premature deaths in 2015 worldwide, placing PM2.5 in the fifth place as a risk factor for global mortality in the Global Burden of Disease, Injuries, and Risk Factors Study (GBD) 2015 [2,3]. Other studies assessing the effect of air pollution levels on mortality have shown that in Europe, in 2019, around 307,000 premature deaths were caused by chronic exposure to fine particulate matter PM10 and PM2.5. In addition, approximately 40,400 premature deaths have been attributed to chronic exposure to nitrogen dioxide (NO_2_) [4]. While emissions of the key air pollutants and their concentrations in ambient air decreased significantly in the last two years in Europe, air quality remains low in many regions. Research shows that in the case of PM2.5, the area of Eastern Europe, including Polish cities, is significantly burdened [4]. According to the World Bank Group, 36 of the 50 most polluted cities in the European Union are located in Poland. In Poland, increased levels of air pollution are associated with a higher incidence of respiratory diseases, including asthma and lung cancer, as well as with increased mortality due to general and respiratory diseases [5]. The results of the available studies indicate that the greatest health risks may be in the countries with the largest populations. However, in relative terms of the number of life years lost per 100,000 inhabitants, the highest relative risk is observed in the countries of Central and Eastern Europe for PM2.5 and in the countries of Central and Southern Europe for NO_2_ [6]. As mentioned, exposure to major air pollutants is associated with increased mortality due to lung cancer and other diseases [7]. Lung cancer is the second most common cancer, accounting for 11.4% of an estimated 2.3 million new cancer cases in 2020 and remains the most common type of cancer among men worldwide [8]. The main cause of lung cancer is cigarette smoking [9,10,11,12]. Among other non-tobacco causes, studies indicate occupational exposure, radon, and air pollution [13,14,15,16,17,18]. Cohort studies conducted in nine European countries showed a relationship between the exposure of residents to air pollution by particulate matter and the risk of lung cancer [19,20,21]. The identification of the relationship between risk factors and the occurrence of lung cancer subtypes has become the subject of many studies [22,23,24]. There is evidence of lung cancer histopathological trends related to risk factors and gender. In the case of squamous cell carcinoma, the dominant factors are heavy smoking and male sex, while in non-smoking women the prevalence of adenocarcinoma is predominant [25,26,27,28]. However, the results of the observations concerning the assessment of the relationship between the histological types of lung cancer and fine solid particles of impurities are not unequivocal [21,22]. The key aspect in these studies is the identification of pollution areas in the place of residence, taking into account the type of pollution, their levels, and the exposure time [29]. PM2.5 and PM10 are present in the air simultaneously, as are gases such as nitrogen dioxide (NO_2_) and sulfur dioxide (SO_2_). For this reason, it is important to verify the impact of selected pollutants selectively and their interaction on the health of the inhabitants. The recent study [30] has shown that the risk factor reveals a significant relationship between the occurrence of the squamous cell carcinoma and the level of PM2.5 and PM10 pollution, among other things (risk ratio amounted to 1.006 and 1.028, respectively). A significant relationship was observed between the spatial distribution of that cancer incidence and the level of PM in Podkarpackie Voivodship.

The aim of the present study was to assess the relationship between long-term exposure to PM2.5, PM10, NO_2_, and SO_2_ and the incidence of lung cancer in the squamous subtype in south-eastern Poland. Compared to the previous work [30], the range of carcinogenic factors was extended and the relationships between them were taken into account. This paper attempts to answer the following questions:(1)Does gender affect its incidence?(2)What is the strength of this relationship with respect to each particular component of environmental pollution? If there is a link, does the environmental pollution map match the squamous cell carcinoma map?(3)What is the likelihood of developing a squamous cell carcinoma, given the intensity of the contamination?(4)What is the trend of the incidence of that cancer in the studied period (2004–2014)?

## 2. Materials and Methods

### 2.1. Design

The study consisted of several parts. First, the information on nearly 11,000 patients from Podkarpackie Voivodship diagnosed with lung cancer in the years 2004–2014 was examined. In the next step, data on the concentration of SO_2_, NO_2_, PM2.5, and PM10 pollutants in Podkarpackie Voivodship were collected over a similar period of time. In order to investigate the relationship between the level of concentrations of individual pollutants and the place of residence of patients with lung cancer in the squamous subtype, proprietary pollution maps were applied to the places of residence of patients. The obtained results allowed to identify which air pollutants and combinations of pollutants were responsible for the increased incidence of squamous cell lung cancer, taking into account gender and the place of residence in Podkarpackie Voivodship in 2004–2014.

### 2.2. Lung Cancer Data

The key data source was the database of the incidences of lung cancer from the Epidemiology Department and Podkarpackie Malignant Tumour Registry of the Chopin Clinical Voivodship Hospital in Rzeszów from 2004 to 2014. In the indicated period, 10,993 cases of all types of lung cancer were registered. The study group consisted of 4237 patients with squamous cell lung cancer. The respondents were grouped by sex because men and women have different tolerance to air pollution [1,2] and age, and then the points with the place of residence of patients were marked on the map of Podkarpackie Voivodship.

### 2.3. Pollution Data

The data sources pertaining to pollutions SO_2_, NO_2_, PM10, and PM2.5 were hourly data from the pollution measurement stations located in Podkarpackie Voivodship from the period 2005–2014. From Voivodship Inspectorate of Environmental Protection as well as data from the annual statistical reports on the emission of air pollutants and on the state of purification devices (OS-1) for the years 1995–2014. There were missing data due to changes of locations of measurement stations as well as technical difficulties. Based on the harmonic analysis, the gaps in hourly data were filled in at all measuring stations due to the fairly stable 24-h, weekly, and annual seasonality. Then the average annual values were established. The data from the OS-1 report was used due to the lack of readings from measuring stations from 1995–2004 after prior aggregation to the level of the local administrative unit. An autoregressive model with an exogenous variable (ARX) [31] was developed for each station, adopting the results from the OS-1 reports for each territorial unit as the explanatory variable.

### 2.4. Data Analysis

A recent epidemiological study by the Harvard School of Public Health found the effects of short- and long-term exposure to pollution. However, these relationships have not been fully elucidated due to different epidemiological methodologies and exposure errors. Hence, new models are proposed to more efficiently evaluate data on short- and long-term human exposure [32]. To analyze the data and find answers to the questions posed, the risk ratio was used as well as a number of statistical methods, i.e., the pollution model, inverse distance weighted interpolation, PCA, and ordered response model.

One of the basic statistics for the analysis of cohort data is the risk ratio (RR). It is defined as the ratio of cancer incidence in the distinguished group to cancer incidence in the control group [33]. The distinguished group is considered to be people living in an area with a pollution level above the voivodship’s average value for a given type of pollution, i.e., PM2.5, PM10, NO_2_, and SO_2_. Using the approach in the work of Szklo and Nieto [34], the confidence interval for the risk ratio was determined and a test was performed whether the risk ratio differs significantly from the value of 1, which means no differences between the distinguished group and the control group.

Since there are no pollution maps for Podkarpackie Voivodship before 2014, a pollution model (covering several methods) was used to develop pollutions maps for 1995–2014. We used a new pollution model that integrates dispersed information sources and was developed. More details about modelling pollutions can be found in the previous article [30].

Using the inverse distance weighted (IDW) interpolation method, the level of pollution in 1995–2014 was estimated for points for which data from other sources were not available [35].

Principal component analysis (PCA) is a data mining technique to visualize more or less initially correlated data, after having clustered them in a reduced space of new variables with the main features to revealing hidden structure. What is important to underscore is that PCA is sensitive to the presence of outliers and therefore also to the presence of gross errors in the datasets. Therefore, a number of tests have been conducted to reveal the sample adequacy and sphericity of correlation between variables. The Kaiser–Meyer–Olkin (KMO) [36] measure of sampling adequacy tests whether the partial correlations among variables are small. As far as the data of our model are concerned, the KMO measure amounts to 0.581. This value is weak but yet acceptable. KMO values greater than 0.8 can be considered good, while this value less than 0.5 requires remedial action. Bartlett’s test of sphericity tests whether the correlation matrix is an identity matrix, which would indicate that the factor model is inappropriate [37]. Bartlett’s test of sphericity rejects null hypothesis with a chi-square value of approximately 355,476 and a significant level of 0.000. An identity correlation matrix means that variables are unrelated and not ideal for factor analysis, suggesting a high level of correlation among variables. As it follows from many studies (e.g., [38]), time–space distribution of particulates is expected to follow a power law (PL). The cited publication found a PL in the case of the time–space weather components such as wind speed, temperature, and humidity. In this paper, we applied a recent test of Gabaix [39] based on the PL known in theory of information to stand as a generalized form of a Gaussian distribution [40]. The Gabaix test rejected the presence of heavy tailed queue in favour of the normality assumption. Nevertheless, taking into account the fact that a PL is a converging law, this result does not prevent from higher variance values as it is the case in this study. Next, data linear interpolation or extrapolation may be to a certain extent the cause of the change of data structure from a PL to a Gaussian structure. It is worthy to recall that in the case of a pure PL, variance and average will converge to infinity. Reassuming and following the above test outputs, it is therefore possible to initiate a PCA on these data.

We have two different samples under analysis, each comprising one response variable (the squamous cancer incidence in the case of women or men) and the following environmental pollution variables:(1)Year_diagnosis: year of cancer diagnosis;(2)Siks_km^2^: number of diagnosed persons with cancer per squared km;(3)SO_2_: sulphur dioxide;(4)NO_2_: nitrogen dioxide;(5)PM2.5: particulate matter of aerodynamic diameter less than 2.5 μm;(6)PM10: particulate matter of aerodynamic diameter less than 10 μm;(7)SO_2_NO_2_: interaction between SO_2_ and NO_2_;(8)SO_2_PM2.5: interaction between SO_2_ and PM2.5;(9)SO_2_PM10: interaction between SO_2_ and PM10;(10)NO_2_PM2.5: interaction between NO_2_ and PM2.5;(11)NO_2_PM10: interaction between NO_2_ and PM10;(12)PM2.5PM10: interaction between PM2.5 and PM10.

The interaction variable is an artificial variable that explains the joint impact on cancer contamination of simultaneous inhalation of given particles. Above aggregate variables display a multiplicative form. In the presence of synergetic effect, impact of each of the variables is less than the one of the composite variable. Thus, we have two datasets each with 12 variables and 8303 time–space squared km- centroids.

For the studied phenomenon, we used the ordered response model. The model assumes that the observed label depends on some hidden continuous variable that can be modeled on the observable variables. Typically, the maximum likelihood method is used [41]. As a response variable, various possible states were taken depending on the number of cancer apparition in a given centroid during the analysed period.

Finally, the spatial maps of environmental pollution (1995 to 2014) were confronted with the spatial map of the cancer incidence (from 2004 to 2014) in Podkarpackie Voivodship measured by 8303 centroids of one square km. At different dates, the cancer may appear in the same centroid but in different environmental conditions. Thus, time series-related cancer incidence was analysed along with corresponding geographical centroids taking into account the gender affected by the disease. As it follows from theory, the model proposed requires a contamination process associated with a long period of exposure. Taking into account the availability of data we retained at least around 10 years of inhalations of NO_2_ and PM10 for all 8303 centroids. This suggests that a person diagnosed with cancer in 2004 (the first year of the study, which ended in 2014) was exposed in 1995–2004. The highest period of exposure is 20 years, from 1995 to 2014. Thus, the process of contamination is a function of multiplicatively aggregated yearly data of pollutant inhalation. The idea behind this aggregation is the hypothesis that each yearly pollutant acts as production factor of a multiplicative function with respect to cancer onset. Nevertheless, we simplified the aggregation procedure and supposed that all pollutants in each year have had the same weight on impacting on the cancer onset. Next, new variables from different particles have been created through a multiplicative form to measure synergetic effect between them. The aggregate variable PM10NO_2_, from variables PM10 or NO_2_ may serve as an illustrative example.

## 3. Results

The maps below show changes in the spatial distribution of pollution in Podkarpackie Voivodship area in the period 1995–2014 (Figure 1).

In the period of 1995–2014, the level of SO_2_ pollutants increased significantly from 4.17 to 4.77 µg/m^3^. The changes are especially in the western part of the region. The average NO_2_ level dropped from 2.29 µg/m^3^ until 2002, when the average NO_2_ level was 1.57 µg/m^3^, and then it started to increase to 2.23 µg/m^3^ in 2014. The emission of PM10 pollutants systematically decreased in the analyzed period from 19.48 to 16.91 µg/m^3^. The opposite trend was observed for PM2.5—an increase from 12.08 to 12.63 µg/m^3^. PM2.5 and PM10 pollution maps are available in our previous publication [30].

In order to calculate the risk ratio, the region was divided into an area with pollution below the average and above the average value in the voivodship, respectively, PM2.5, PM10, NO_2_, and SO_2_. Then, the number of people living in these areas was estimated, broken down by gender and two age cohorts: under the age of 75 and others. The division into such groups is related to the age at the peak of cancer incidence in the available database. The tables below present the results of the risk ratio broken down by gender and two age cohorts for people living in areas above the average pollution level in the voivodship.

It turns out that the group of people under the age of 75, with the age of cancer diagnosis starting from the age of 28, is particularly exposed to the carcinogenic effects of air pollution, regardless of gender. The risk ratio is significantly greater than one at the significance level of *p* < 0.0001 in all cases in the group of women and the entire population (Table 1). In the group of men, RR indicates an increased risk of developing the disease, but the results are not significant (*p*-value > 0.1) (Table 2). In the group of people aged 75 and over, the influence of air pollution on the increased risk of developing the disease is visible only among women. In the group of women, the risk ratio is significantly greater than one (*p*-value < 0.0001). The results are not significant in the group of men and the general population. However, it should be taken into account that the incidence in this age group is also influenced by factors cumulating with age.

Next, we applied the principal component analysis (PCA) [42]. As it is shown in Table 3 and Table 4 of the PCA outputs, the obtained two principal component account for around 74% of all information contained in the data both for women and for men cases. If we add the third factor (time), they together account for 95% of all the variability in the original data. Since each of the three sets of data (three axes) are independent of the others, it would therefore be interesting to find a relation between each set of variables and the factor that represents them from the correlation point of view. As displayed in Table 3 for the case of women cancer, the main factor (component 1) is highly correlated with the following variables: NO_2_PM2.5, SO_2_NO_2_, NO_2_, SO_2_, NO_2_PM10, and the second factor (component 2) with: SO_2_PM2.5, PM2.5PM10, PM2.5, PM10, and SO_2_PM10. The third factor (component 3) is strongly correlated with the time of diagnosis. Thus, the first factor can be renamed the nitrogen and sulphur oxides axis, the second factor could be renamed the atmospheric particulate matter (PM) and the third factor as the one of cancer diagnosis over time. Since the above outputs in Table 3 are quite identical in the case of men cancer, we obtained the same conclusions and the same names of axes. In both samples, we obtained the very weak correlations of the variable “Siks_km^2^” with all axes, which reflects the fact that this variable is very close to the gravity center.

Since the distance from this center indicates for variables the level of correlation with the axes, it is likely that the synergetic joint effect of particles NO_2_ and PM2.5 (i.e., the variable NO_2_PM2.5) (see Table 1) should play a dominant influence in the formation of the axis 1, for example.

We noticed the strongest influence of the nitrogen dioxide NO_2_ in the creation of the first principal component followed by the sulfur dioxide SO_2_. The particulates PM10 and PM2.5 remain more correlated with the second principal axis. Since the correlation of the variable “number of sicks” (Sicks_km^2^) with all axes remains very weak but close to the origin, one can then consider them grossly as the center of gravity of principal components and analyse their relationships with the rest of the variables. For instance, we observe from Table 3 and Table 4 on the third axis that the time of diagnosis (from 2004 to 2014) is positively correlated with the number of sick women. On the contrary, this relationship is negative in the case of sick men, suggesting that their number decreases over time. In conclusion, we can expect to obtain in the next paragraphs of this paper the model estimates that display approximately that the relationships conform to those revealed through PCA outputs visualization.

In the case of women cancer (see Table 5), the main factor seems to be the joint inhalation of the nitrogen dioxide NO_2_ and the particulate PM10, that is the variable NO_2_PM10. In the case of men cancer (see Table 5) we notice that the main factor seems to be a synergetic effect caused by the joint inhalation of the nitrogen dioxide NO_2_ and the sulfur dioxide SO_2_, that is the variable NO_2_ SO_2_ plus the time with a negative sign on the estimate, which means that over time we averagely had a diminution of sick men of squamous cancer in Podkarpackie Voivodship.

Except for the variable “Year_diagnosis”, whose data have not been normalized, the remaining data have been initially standardized and estimates in the table above are related to them.

Table 6 and Table 7 below present the statistical average risk evaluation of the probability of being contaminated in the case of an inhabitant living in a given Podkarpackie centroid in a given year over the period of 2004–2014. The first row of both tables explains the possible number of contaminated people. The second row presents the factor of contamination, the third row the average marginal effect that is the relative odds of the cancer onset due to the ceteris paribus particles, and/or gas level change in each centroid. In the case of women (Table 6), an increase in one sigma_x (see the fourth row of Table 6), which is a quantity of 5.16853 × 10^22^ one µg/m^3^ of NO_2_ times PM10 in μms measure units leads to a reduction of the odds of not catching cancer by 9‰ (−0.008913), ceteris paribus. We need to comment on the obtained huge value of the standard deviation of PM10NO_2_. This is due to the multiplicative character of the initial model for the preparation of the present variable input data (in this case PM10 and NO_2_). As it follows from the above comment in Section 2.4, the big value of the standard error is related to this multiplicative aggregation of data from a large time–space range displaying higher value variability. It is worthy to inform the readers that additive aggregation of yearly value pollutants led to meaningless outputs. The large value of the standard deviation depends too on the used units and the time–space scale. As far as the interpretation of the results is concerned, we can say that in any case, getting sick requires huge quantities of pollutants if only these have to remain the alone cause of contamination. The last row of Table 6 explains the cancer onset average relative risk in Podkarpackie Voivodship. The average probability of not catching cancer is around 93%, while the one that only one person living in a given centroid is around 6% over the period 2004–2014.

The same statistical interpretation applies in the case of Table 7. Nevertheless, our conclusions should be understood in the context of the model precision. As shown in the comments below Table 5, we see that the number of the correct prediction cases is around 93% in the case of the women cancer model. This precision is reduced to only 61% in the case of men. We recall that the number of correct prediction is obtained by rounding up/down from 50% probability score of each case.

## 4. Discussion

One of the greatest plagues of our era is air pollution, not only because of its impact on climate change, but also for public and individual health, due to the increasing morbidity and mortality from various diseases [43]. There are many pollutants that are major contributors to human disease. Carcinogens found in airborne particles may interact directly with the cells of the lungs causing their inflammation. In turn, chronic inflammation may also play a role in the development of lung cancer, with neoplastic lesions appearing in response to exposure to irritants and repeated damage [44]. There are many pollutants in the air that, in high concentrations, make you susceptible to many diseases, including various types of cancer, including the lungs. Research on the influence of air pollution on the formation of lung cancer has been conducted for nearly 50 years. At first, it was postulated that the main cause of lung cancer may be long-term exposure to air pollutants. At that time, however, the actual risk was estimated to be much lower than that associated with smoking [45]. More recent reports confirm the relationship between air pollution and mortality from lung cancer [46,47,48,49]. Age is also an important factor influencing the incidence [50,51,52,53].

The conducted research has shown that people up to 75 years of age, with the age of tumor diagnosis starting from the age of 28, are particularly vulnerable to carcinogenic effects of air pollution, regardless of gender. In turn, in the group of people aged 75 and over, the influence of air pollution on the increased risk of developing the disease is visible only among women. However, it should be taken into account that the incidence in this age group is also influenced by factors cumulating with age. There is ample evidence of a strong association between PM2.5 and PM10 with various respiratory diseases [54,55,56]. Based on a meta-analysis of 18 studies, a correlation was observed between PM2.5 and PM10 particles and the incidence and mortality of lung cancer, with PM2.5 being considered the most important pollutant [23]. The present studies showed a secondary relationship of the selectively treated PM2.5 and PM10 particulates. A greater impact on the disease was identified in the combination of the indicated pollutants with NO_2_. Wang et al. suggested that the carcinogenic effects of PM2.5 particles differ according to gender as well as the environment in which people live, i.e., in the countryside or in the city. The analyzes showed that women had a significant risk of developing lung cancer in correlation with exposure to PM2.5 [57]. On the other hand, other studies indicate that PM10 particles play a greater role in the incidence of lung cancer in women than in men. This shows that gender differences can cause different responses to the same pollution [58]. The same studies show that PM10 pollution affects the type of lung cancer differently [51]. The research also showed a difference between sex and pollution.

We noticed in the case of squamous cell lung cancer in women (Table 3) that the main factor seems to be the joint inhalation of NO_2_ dioxide and PM10 particulate matter, the variable NO_2_MP10. In the case of squamous cell lung cancer in men (Table 3), we note that the main factor seems to be the joint inhalation of NO_2_ and SO_2_ dioxide, i.e., the variable NO_2_ SO_2_. The irritating effect of nitric oxide on the respiratory system has already been confirmed in other reports [59]. Penetrating deep into the lungs, into the respiratory tract, it causes diseases of the respiratory system, and directly causes coughing, wheezing, shortness of breath, bronchospasm and even pulmonary edema when inhaled in large amounts. These studies confirm the harmfulness of NO_2_ dioxide, especially when it is combined with other pollutants simultaneously present in the environment. The main health problems related to SO_2_ emissions in industrialized areas are respiratory tract irritation, bronchitis, mucus production, and bronchospasm as it is an irritating sensory agent and penetrates deep into the lungs converted into bisulfite and interacts with sensory receptors causing bronchospasm [60]. The analysis of the collected data showed a significant impact of SO_2_ concentration on the incidence of lung cancer in the squamous subtype, considered both selectively and in combination with other compounds. Increased concentrations of SO_2_ dioxide and its combination with NO_2_ seem to play the role of primary carcinogens, while the combination of SO_2_ dioxide with PM2.5 and PM10 particles may play a secondary role and influence the formation of a lung tumor in the analyzed histological subtype. The influence of pollutants on lung cancer is most often considered selectively [61]. It turns out, however, that the toxicity of several air pollutants assessed together may also cause various neoplasms in the long run [62].

The results of these studies confirm the toxicity of various combinations of pollutant compounds, in particular the variable NO_2_PM10 and NO_2_ SO_2_. Based on the available data, the analyzes carried out showed a decline in the incidence of lung cancer in the squamous cell subtype in Podkarpackie Voivodship in the case of men. However, taking into account the significant carcinogenic effect of the NO_2_ SO_2_ variable and the data on the increasing concentration of NO_2_ and the particular increase in SO_2_ in the western part of Podkarpackie Voivodship, the probability of developing lung cancer in the squamous subtype in men increases in this area.

## 5. Conclusions

Cancer in women and in men seems to depend in particular on the variables NO_2_PM10 and NO_2_ SO_2_, respectively. The trend of contamination among women—in particular due to the NO_2_ PM10 synergy effect—remained stable over the period analysed, 2004–2014. In the case of men, this trend is due in particular to the NO_2_ SO_2_ synergistic effect and was decreasing during the same period. Even if this conclusion seems to correspond to intuition, more thorough research is desirable to understand the causes. Nitrogen dioxide plays the leading role, which once composed with the other elements it becomes more persistent in explaining the appearance of cancers and could constitute the main cause of squamous cancer. It is recommended to intensify prophylactic measures towards lung cancer in the areas of Podkarpackie Voivodship with increased NO_2_ and SO_2_ concentrations, especially in its western region. Poland needs more multi-city studies to consistently track the disease burden from air pollution.

## Figures and Tables

**Figure 1 ijerph-19-11598-f001:**
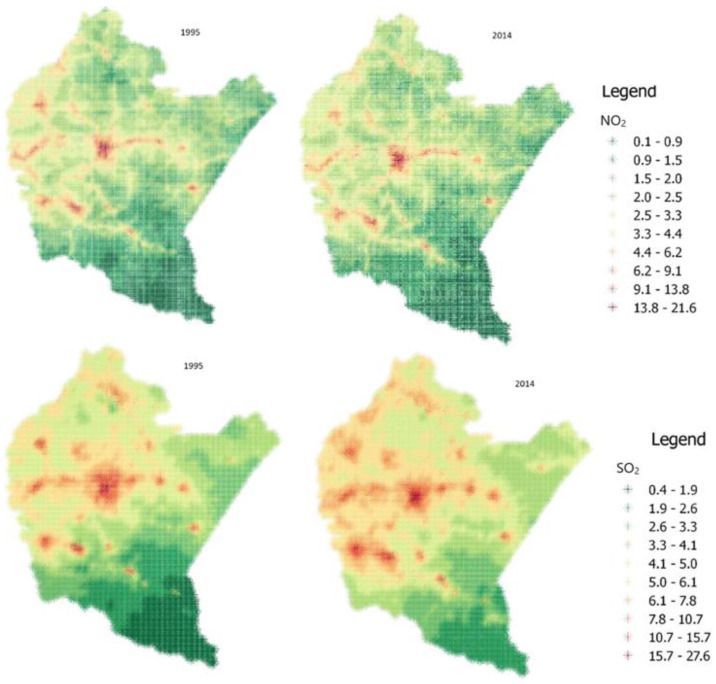
Maps of NO_2_ (µg/m^3^) and SO_2_ (µg/m^3^) pollutants in 1995 and 2014.

**Table 1 ijerph-19-11598-t001:** Risk ratio for the population below the median age (<75) (with 0.95 confidence interval).

	SO_2_	NO_2_	PM2.5	PM10
Women	1.1326 (1.037–1.237)	1.1548 (1.058–1.261)	1.1201 (1.025–1.225)	1.1476 (1.049–1.256)
Men	1.0136 (0.963–1.067)	1.0202 (0.969–1.074)	1.0227 (0.971–1.077)	1.0360 (0.983–1.092)
Total	1.0360 (0.991–1.083)	1.0462 (1.001–1.094)	1.0397 (0.994–1.087)	1.0563 (1.009–1.105)

**Table 2 ijerph-19-11598-t002:** Risk ratio for populations above the median (75+).

	SO_2_	NO_2_	PM2.5	PM10
Women	1.0949 (1.000–1.199)	1.1022 (1.006–1.207)	1.1529 (1.053–1.262)	1.1742 (1.072–1.287)
Men	0.9245 (0.883–0.968)	0.9182 (0.877–0.961)	0.9412 (0.899–0.986)	0.9613 (0.918–1.007)
Total	0.9648 (0.926–1.006)	0.9618 (0.923–1.003)	0.9895 (0.949–1.032)	1.0092 (0.968–1.053)

**Table 3 ijerph-19-11598-t003:** Correlation between axis and variables (women squamous cell carcinoma).

	Rescaled		
	Component		
	1	2	3
NO_2_PM2.5	0.995	0.033	0.016
SO_2_ NO_2_	0.993	0.016	0.014
NO_2_	0.986	0.015	0.021
SO_2_	0.980	0.039	0.023
NO_2_PM10	0.976	0.017	0.012
Siks_km^2^	0.035756	−0.001907	0.0272667
SO_2_PM2.5	−0.019	0.967	−0.002
PM2.5PM10	−0.023	0.963	−0.002
PM2.5	−0.027	0.901	−0.001
PM10	0.003	0.870	0.013
SO_2_PM10	0.129	0.750	0.007
Year diagnosis	0.008	0.023	0.9997061

**Table 4 ijerph-19-11598-t004:** Correlation between axes and variables (men squamous cell carcinoma).

	Rescaled		
	Component		
	1	2	3
NO_2_PM2.5	0.994	0.030	−0.025
SO_2_ NO_2_	0.993	0.013	−0.028
NO_2_	0.987	0.012	−0.021
SO_2_	0.980	0.036	−0.017
NO_2_PM10	0.975	0.015	−0.029
SO_2_PM2.5	−0.015	0.966	0.045
PM2.5PM10	−0.019	0.962	0.045
PM2.5	−0.023	0.900	0.043
PM10	0.007	0.869	0.054
SO_2_PM10	0.132	0.749	0.037
Year diagnosis	0.051	−0.025	0.9984106
Siks_km^2^	0.0061416	−0.008859	−0.044846

**Table 5 ijerph-19-11598-t005:** Correlation between air pollution and the incidence of squamous cell lung cancer by gender.

	Year Diagnosis	SO_2_ NO_2_	NO_2_PM10
Women			0.0223
*p*-value			<0.001
Men	−0.0175	0.0104	
*p*-value	<0.001	0.0027	

Comments: Number of correct predictions cases for women = 7746 (93.3%); Number of correct. predictions cases for men = 5062 (61.0%).

**Table 6 ijerph-19-11598-t006:** Women squamous cancer: factors and risks.

Number of Contaminated in a Centroid	0	1	2	3	4+
Factor of Contamination	NO_2_PM10	NO_2_PM10	NO_2_PM10	NO_2_PM10	NO_2_PM10
**Average marginal effect**	−0.0089	0.0086	0.0003	0.000019	0.000011
**One Sigma_x: one unit of standard error of a concentration of 1 µg m^−3^ of NO_2_** **times PM10 in µms measure units**	5.17 × 10^22^	5.17 × 10^22^	5.17 × 10^22^	5.17 × 10^22^	5.17 × 10^22^
**Cancer onset average relative risk in Podkarpackie**	0.9328	0.0625	0.0043	0.0002	0.0001

Average marginal effects (the relative odds of the cancer onset due to the particles and/or gas level change in EACH CENTROID with respect to others alternative).

**Table 7 ijerph-19-11598-t007:** Men squamous cancer: factors and risks.

Number of Contaminated in a Centroid	0		1		2		3		4		5		6+	
Factor of Contamination	SO_2_ NO_2_	Time	SO_2_ NO_2_	Time	SO_2_ NO_2_	Time	SO_2_ NO_2_	Time	SO_2_NO_2_	Time	SO_2_NO_2_	Time	SO_2_ NO_2_	Time
**Average marginal effect**	0	0	−0.000781	0.001322	0.000497	−0.000841	0.000207	−0.000349	0.000052	0	0.000017	−0.000028	0.000009	−0.000016
**One Sigma_x: one unit of standard error of a concentration of 1 µg m^−3^ of** **NO_2_** **times PM10 in µm s measure units**	8.636 × 10^47^		8.636 × 10^47^		8.636 × 10^47^		8.636 × 10^47^		8.636 × 10^47^		8.636 × 10^47^		8.636 × 10^47^	
**Cancer onset average relative risk in Podkarpackie**	0.609506025		0.356412517		0.023710488		0.007954226		0.001691009		0.000483947		0.000241788	

Average marginal effects (the relative odds of the cancer onset due to the particles and/or gas level change in EACH CENTROID with respect to others alternative).

## Data Availability

Not applicable.

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
