# Peer review of "Association between Air Pollution and Squamous Cell Lung Cancer in South-Eastern Poland"

_ijerph, 2022, doi:10.3390/ijerph191811598_

Round 1

Reviewer 1 Report

This is an interesting study that assessed the relationship between long-term exposure to PM2.5, PM10, and NO2 based on ten years of data collection. the model used in the study is acceptable, and the analysis and discussions of the results are reasonable. However, to further improve the work, I only have a few comments/suggestions for the authors to consider:

1. The main problem is the data analysis on the impacts of SO2, NO2, PM10, and PM2.5. As we know, both SO2 and NO2 are the primary component of PM10 and PM2.5. Thus, the influences of PM10 and PM2.5 on lung cancer may include the effectiveness of SO2 and NO2.

2. Double-check the format of NO2 , PM10, pm2.5, and SO2 through the main text.

3. Line 148: Please briefly describe the pollution model for readers' quick understanding.

4. Line 203: figure 1, it seems there were no changes of NO2 from 1995 to 2014. Please add the unit to figure 1. If possible, please add more figures in the main test, e.g., 1995,2000,2005,2010. Moreover, it is important to show data in 2004, the first year of your study.

Author Response

Dear Reviewer,

In reference to your suggestions, I am sending replies to the left comments:

  1. The main problem is the data analysis on the impacts of SO2, NO2, PM10, and PM2.5. As we know, both SO2 and NO2 are the primary component of PM10 and PM2.5. Thus, the influences of PM10 and PM2.5 on lung cancer may include the effectiveness of SO2 and NO2.

This study attempts to analyze the impact of selected air pollutants. The authors of the study are aware that SO2 and NO2 are the primary component of PM10 and PM2.5. However, our analyzes focused on the collected unit pollutant data, which reliably and reliably determine the concentration of individual pollutants.

  1. Double-check the format of NO2 , PM10, pm2.5, and SO2 through the main text.

We made corrections to the text

  1. Line 148: Please briefly describe the pollution model for readers' quick understanding.

A short description has been added.

  1. Line 203: figure 1, it seems there were no changes of NO2 from 1995 to 2014. Please add the unit to figure 1. If possible, please add more figures in the main test, e.g., 1995,2000,2005,2010. Moreover, it is important to show data in 2004, the first year of your study.

The authors of the work did not take into account the visualization for individual years e.g. 2000, 2005, 2010, as the aim of the study was to assess the long-term exposure to pollution, i.e. a ten-year period.

With Kind Regards,

Michalina Czarnota

Reviewer 2 Report

In the manuscript entitled “Association between air pollution and squamous cell lung cancer in South-Eastern Poland” the authors were presented the connection between airborne pollutants and the incidence of squamous cell lung cancer. Overall, the work is written appropriately and is easy to read and follow. The paper is carefully thought and performed. All experimental methods and analyses are explained. The authors had a lot of data to review, 4,237 patients The data presented is significant in convincing many people of the impact of pollution on the human body. As the authors wrote: Poland needs more multi-city studies to consistently track the disease burden from air pollution.

However, I have some reservations about the authors:

1.      The abbreviation designations of PM2.5, PM10, SO2 and NO2 are glaring and such a record is a very big oversimplification for a serious magazine (IJERPH). Sulfur dioxide and nitrogen dioxide are written with subscripts, for PM numbers as well. See: https://www.epa.gov/so2-pollution/setting-and-reviewing-standards-control-so2-pollution

Please, correct it.

2.      More comments are highlighted in the text, pdf.

Author Response

Dear Reviewer,

In reference to your suggestions, I am sending replies to the left comments :

  1. The abbreviation designations of PM2.5, PM10, SO2 and NO2 are glaring and such a record is a very big oversimplification for a serious magazine (IJERPH). Sulfur dioxide and nitrogen dioxide are written with subscripts, for PM numbers as well. See: https://www.epa.gov/so2-pollution/setting-and-reviewing-standards-control-so2-pollution

Please, correct it.

We made corrections to the text.

  1. More comments are highlighted in the text, pdf.
  2. a) Insert a point/dot instead of a comma. - We made corrections to the text.
  3. b) There is an error instead the reference. - We made corrections to the text.
  4. c) From line 179 to 183 instead of the letter O, there is a zero. - We made corrections to the text.
  5. d) Use the impersonal form. - We made corrections to the text.
  6. e) The author of this article does not appear in the cited manuscript. Thus, the word "our" should not occur. - There was an error in the text. The correct reference number has been entered
  7. f) Why 75? The division into groups 75 aged is related to the age at the peak of cancer incidence in the available database.
  8. g) Dot instead comma.- We made corrections to the text.
  9. h) Point instead comma. - We made corrections to the text.
  10. i) Does it mean promille? - We made corrections to the text.
  11. j) Why did women have a significant risk of developing lung cancer in correlation with exposure to PM2.5 and PM10? Has it been reported in the literature? - We made corrections to the text.
  12. k) PM10 instead MP10. - We made corrections to the text.
  13. l) In the conclusions, I propose to explicitly answer the questions posed on page 2. What is missing here is a reference to time. - We made corrections to the text.
  14. m) file:///C:/Users/annas/Downloads/2021-10%20Eionet%20report_HRA%20-%20FINAL1%20for%20publication.pdf At this link is the publication to which the authors referred. - We made corrections to the text.

With Kind Regards,

Michalina Czarnota

Author Response

Dear Reviewer,

In reference to your suggestions, I am sending replies to the left comments:

Dear authors, understanding the effect of air pollution on the population is an important topic of research. however, it feels like no attempt to understand the air pollution problem was made. it is most obvious in the attempt to pass a computer variable as a cause for cancer. specifically, NO2SO2 one. please note my specific comments below. Take extra attention to comments 3, 13, 23, and 24. they are all related.

  1. Language corrections that were spotted:

Line 40 and anywhere where relevant (line 72, 99, 117 etc): NO2 - the chemical equation for Nitrogen dioxide is NO2 with a subscript for "2": NO2. Same for SO2: SO2line 47-48: People living in more polluted regions (southern Poland) have a higher level of air pollution - if you live in a more polluted area, you have higher pollution. true for everywhere, not only Poland. My point is, this sentence repeats itself, please rephrase.

line 51-54: However, in relative terms, taking into account, for example, the number of life years lost per 100,000 inhabitants, the highest relative risk is observed in the countries of Central and Eastern Europe for PM2.5, in the countries of Central and Southern Europe for NO2 [7]. - this entire sentence is too long and contains too many commas. please revise. I suggest to use "and" when doing lists and remove "countries", just reference sections of Europe.

Live 164: approximately 355475.911 - what is the point of reporting 3 numbers after the decimal point for such a high number? especially after you write approximately.

Answer: correction has been added into the main text

Line 206: 4.77 g / m3 - should either have ^ or use super script. this is not an accepted

unit. please fix this everywhere it is shown. Answer: correction has been added into the main text

Line 288: -3 of NO2 - keep your units constant. eith Answer: correction has been added into the main text

er /m^3 ( as used above) or m^-3 as implied here. Answer: correction has been added into the main text

Line 288: - what is ms? micro milliseconds? how is that a unit of PM10? Answer: correction has been added into the main text

Line 289: by 9%o - Word (and all other modern software) have a sign for Per-mille: . please use it, and not this weird combination of percent and a circle. Answer: correction has been added into the main text

  1. Line 22: statistical methods, i.e. the pollution model - why is a pollution model also a statistical method?

Answer: correction has been added into the main text

  1. Line 24: depend in particular on the variables NO2PM10 and NO2SO2 respectively - what do these variables mean in the real world? there is no pollutant called NO2PM10 (we already have enough to worry about, no need to add more). please explain, briefly, as this is a summary, what does this parameter mean in the real world? exposure to both pollutant? a synergistic effect? variations of this comment will return in the relevant sections several times.

Answer:  extensive explanations have been added into the main text

  1. line 50-51: The results of the available studies indicate that the greatest health risks may be in the countries with the largest populations - which results? there is no reference here. also, it sounds like a bias to me. more population, more people exposed. doesn't mean the air pollution is the worst. please add references and clarify. Answer: correction has been added into the main text
  2. line 112-113: The respondents were grouped by sex because men and women have different tolerance to air pollution - I do not understand why this bias was introduced. Especially considering this is still being disputed and discussed:

https://www.ncbi.nlm.nih.gov/pmc/articles/PMC7464921/#:~:text=No%20gender%20differences%20in%20the,m3%20increase%20of%20PM2.

https://www.frontiersin.org/articles/10.3389/fpubh.2019.00375/full

(this study indeed found women are more effected, but by a negligible amount 1.0-1.05 or 1.01-1.07).

furthermore, you provide results (briefly, but they are there) for the total population. please revise.

  1. Line 117-121: The data sources pertaining to pollutions SO2, NO2, PM10 and PM2,5 were hourly data from the pollution measurement stations located in Podkarpackie Voivodship from the period 2005-2014 from Voivodship Inspectorate of Environmental Protection as well as data from the annual statistical reports on the emission of air pollutants and on the state of purification devices (OS-1) for the years 1995-2014. - too long a sentence. split into two at least, especially since you have two data sources. please clarify which data you got from where and include references or links to databases/websites if possible.
  2. line 148: [Error! Book-148 mark not defined.] - reference broken
  3. Line 155: variables with the main to revealing hidden structure. - the main what? the sentence is unclear, please clarify

Answer: correction has been added into the main text

  1. Line 160-161: the Kaiser-Meyer-Olkin Measure amounts to 0.581. This value is weak but yet acceptable - please provide a reference to it s acceptable values.

Answer: Detailed explanations has been added into the main text

  1. Line 161-163: Bartlett's test of sphericity tests whether the correlation matrix is an identity matrix, which would indicate that the factor model is inappropriate [39]. Bartlett's Test of Sphericity rejects HO - I don't understand why this test is needed at all. the entire part about this test need clarifying. also, explain what is HO.

Answer: The rationale of proposing the test has been presented and othersdetailed explanations has been added into the main text

  1. Live 164-165: high level of correlation among variables - high correlation between variables is bad for any study. are you sure those values are correct?

Answer: yes, correlation of variables dwelling on the same axis is the targeted  result of the PCA as a classification method by its nature. Its worthy to recall that axes are orthogonal to each other.

  1. Line 174: Siks_km2: number of contaminated persons per squared km - what is contaminated persons? people who were diagnosed with cancer? or people exposed to air pollution? which is just the number of people living in the km2

?

Answer: number of contaminated persons per squared km. It means people who were diagnosed with cancer

  1. Line 185: The interaction variable is an artificial variable - how is the interaction defined? is it a sum? A multiplication? a more complicated equation? maybe GAM? if you do not provide a description of it, no one else can ever reproduce your results and do a follow up study

Answer:  extensive explanations have been added into the main text

  1. Line 203: Figure 1 - issues with the figure: the color legend is unclear. NO2 and SO2 has units. please use

them in the figure. is this yearly average? please state so.

  1. Line 212-213: Risk Ratio, the region was divided into an area with pollution below the average and above the average value in the voivodship, respectively - please use the same terms in the methods section for RR: control and distinguished . 16. Line 223: is particularly exposed to the carcinogenic effects of air pollution - can someone be exposed to only one aspect of air pollution? please clarify.
  2. Line 224: The Risk Ratio is significantly greater than 1 - what is significantly greater? 1.5? 5? 100? please provide a reference and clarify. Answer: correction has been added into the main text

same for line 229.

  1. Line 241: variables: N02PM2.5, S02N02, NO2, SO2, N02PM10 - is there any reason to have both NO2, SO2 and NO2SO2? aren't they inherently correlated and biased? please clarify

Answer:  we considered that inhalation of NO2 or SO2 should have a different effect as the simultaneous inhalation of NO2 and SO2. The latter may produce systemic( synergetic) effect in a given ecological environment. Idem with other pollutants. An extensive explanations have been added into the main text.

  1. Line 255 + 256: the dioxides NO2 + the dioxides SO2. - NO2 is NO2. you are saying here Dioxide nitrogen dioxide. that doesn't make sense. the same for SO2 in the line below.
  2. Line 256: The particulates PM same as comment 19 above, particulates particle matter?

please revise

  1. Line 288: that is a quantity of 5.16853E+22 - this is a huge and impossible increase for even a combination of pollutants. please re check your values and calculation. I cannot verify your numbers, as I do not know the equation you used between the variables. what does NO2PM10 composed of? what does this huge number mean in the real world? please clarify

Answer: no error could be detected. Obviously, this standard deviation value is huge in the scale (units) in which the particles are explained. We had to add some additional clarifications that were missing in the previous text. By nature, the structure of variations in the amount of each particle in a given centroid over a period of one year is irregular and large. Then, if we consider that each variable is an aggregate of a multiplicative combination of annual amounts of particles (over a period of 10 to 20 years with respect to the time of exposure to pollutants), one can understand the origin of these enormous values. Please find additional arguments in the main text.

  1. Line 293: Table 6: 5.16853E - E with nothing afterwards is not a number. you should always write a number. 1E1 is just 1. 1E0 is 0.1. etc. there is no need to report 5 numbers after the decimals. please follow the conventions on significant digits.

Answer: the problem has been fixed

  1. Line 367: variable NO2PM10 and NO2SO2 - again, what does this variable mean in the real world?

Answer: these may explain synergetic effect resulting from a simultaneous exposure to both particulates. It may differ from an isolated exposure to both particulates in different time.

  1. Line 370: significant carcinogenic effect of the NO2SO2 variable - a variable cannot be carcinogenic. it's a computer variable. a statistical parameter you produced. NO2 and SO2 can be carcinogenic. please revise.

Answer: the problem has been fixed

With Kind Regards,

Michalina Czarnota